# The role of SNAP and WIC participation and racialized legal status in U.S. farmworker health

**Briana E. Rockler**[1]*, **Stephanie K. Grutzmacher**[2☉], **Jonathan Garcia**[2☉], **Ellen Smit**[2☉]

**1** Department of Public Health and Environmental Studies, College of Arts and Sciences, University of Wisconsin–Eau Claire, Eau Claire, Wisconsin, United States of America, **2** School of Biological and Population Health Sciences, College of Public Health and Human Sciences, Oregon State University, Corvallis, Oregon, United States of America

☉ These authors contributed equally to this work.
* rocklebe@uwec.edu

## Abstract

### Background

Policies that restrict access to and use of the Supplemental Nutrition Assistance Program (SNAP) and Special Supplemental Nutrition Assistance Program for Women, Infants, and Children (WIC) by legal status may disproportionately disadvantage particular racial and ethnic groups. While immigrant legal status, race, and ethnicity are recognized as independent social determinants of health, studies examining the extent to which legal status structures racial and ethnic health disparities are limited. Research is needed to identify factors that mitigate disparate health outcomes, such as SNAP and WIC.

### Methods

Cross-sectional data from the 2009/2010 National Agricultural Workers Survey (N = 3,961) were analyzed. Chi-square tests and logistic regressions examined associations among self-reported health, race, ethnicity, legal status, and SNAP/WIC participation.

### Results

Farmworkers reporting excellent or good health were more likely to be non-Hispanic White, U.S. citizen, aged 18–25, single, male, educated beyond primary school, living above the poverty level, without chronic health conditions, and located in the Midwest. Hispanic farmworkers had lower odds of reporting excellent or good health (OR 0.27, 95% CI 0.12–0.62). Among SNAP/WIC participants, Hispanic farmworkers had higher odds of reporting excellent or good health (OR 6.74, 95% CI 1.54–29.57) compared to non-Hispanic White farmworkers. There was no significant association between self-reported health and legal status.

### Discussion

This study complements the extant literature showing racial and ethnic health disparities among the U.S. farmworker population. Results provide valuable insight on the health-

**Data Availability Statement:** Data from this study are held in a public repository that can be located here: https://www.dol.gov/agencies/eta/national-agricultural-workers-survey.

**Funding:** The author(s) received no specific funding for this work.

**Competing interests:** The authors have declared that no competing interests exist.

protective potential of programs like SNAP and WIC, particularly among Hispanic farmworkers, who may be both less likely to be eligible and more hesitant to participate. These findings underscore the need to expand U.S. farmworkers' eligibility and participation in SNAP and WIC.

## Introduction

Recent estimates show a record number of immigrants living in the U.S., comprising 13.7% of the population. While most are in the country legally, unauthorized immigrants comprise almost a quarter (23%) of the U.S. foreign-born population [1] and 4.8% of the workforce [2]. Seventy-four percent of unauthorized immigrants work in essential infrastructure jobs [3], yet many lack access to fundamental rights and resources. Unauthorized immigrant workers remain subject to workplace discrimination, have few occupational protections, face ongoing detention and deportation threats, and hold little power in the workplace [4]. Legal status also governs citizens' access to rights [4] and social resources through exclusion, stigmatization, and discrimination [5]. As such, legal status structures the differential health risks encountered by immigrants (e.g., stress, occupational conditions), resources available to manage those risks (e.g., income, education), and access to health-promoting resources (e.g., public assistance, health care) [6, 7]. Simultaneously, significant health disparities persist among marginalized racial and ethnic groups in the U.S., including higher rates of chronic disease and premature death than the non-Hispanic White population [8].

Hierarchies of legal stratification may shape the social, political, and economic conditions that determine health outcomes and health inequities. In recognition of this influence, racialized legal status (RLS) has been identified as an emerging social determinant of health. RLS refers to the impact of ostensibly race-neutral legal stratifications that disproportionately disadvantage minoritized racial and ethnic groups [9]. While immigrant legal status, race, and ethnicity are recognized as independent social determinants of health [10–13], research examining the extent to which legal status shapes racial/ethnic health disparities is limited. To better understand RLS as a health determinant, research must focus on the intersectionality between legal status, race, health, and the mechanisms through which legal status determines health, such as participation in public assistance programs and resources.

Two public assistance programs that have demonstrated health benefits for low-income communities [14] are the Supplemental Nutrition Assistance Program (SNAP) and the Special Supplemental Nutrition Assistance Program for Women, Infants, and Children (WIC). Participation in SNAP and WIC may be affected by RLS through federal eligibility guidelines, which limit SNAP participation by legal status. Qualified individuals include U.S. citizens and qualified noncitizens, including children, refugees, asylees, and qualified immigrant adults in the U.S. for at least five years. In addition, SNAP eligibility is contingent upon criteria that vary state to state, including income, assets, age, disability, employment status, and participation time limits. Though there are no eligibility restrictions by legal status for WIC, the program is specific for pregnant and breastfeeding mothers, infants under one year old, and children under five who fall below specified income limits [15].

Nationally, approximately one-quarter of farmworkers are unauthorized, and 83% are Hispanic/Latinx [2]. As the agricultural workforce has a high concentration of unauthorized Hispanic/Latinx immigrants [16], studying U.S. farmworkers may help clarify how RLS affects health. The current study sought to (1) document how race, ethnicity, and legal status are

associated with farmworker health and (2) examine how the use of health-promoting public resources, specifically SNAP or WIC, influence the associations between race, ethnicity, legal status, and health. Results from this study may clarify the role of RLS as a social determinant of farmworker health and provide recommendations regarding SNAP and WIC as programs to improve farmworker health outcomes.

## Conceptual framework

Asad and Clair describe three pathways through which RLS contributes to racial/ethnic health disparities [9]. The primary path is direct, representing the impacts of RLS on the health of any individual who holds discredited legal status, like unauthorized immigrants, through multilevel discrimination and stress, psychosocial consequences of stigma rooted in anti-immigrant rhetoric, and policies that limit access to health-promoting economic and social resources [17]. Two spillover pathways, individual and collateral, demonstrate how RLS affects those with social and cultural proximity to individuals with discredited legal status, such as racial and ethnic in-group members. Individual spillovers affect family members and neighbors through shared proximate risk factors such as stress or forced family separation. Collateral spillovers impact racial/ethnic group members who are misidentified as holding discredited legal status and are subject to comparable discrimination and attempted enforcement as a result. If RLS is a social determinant of health, we expect that the adverse health outcomes associated with legal status impact the entire social and cultural group. For example, policies that restrict access to and use of SNAP and WIC under the auspice of legal status would not only affect unauthorized immigrants, but they would affect racial and ethnic groups with shared social proximity, thus limiting the role that SNAP and WIC play in health promotion among vulnerable communities. As such, a comparison of health across legal strata within a distinct group, like racial and ethnic groups of U.S. farmworkers, may clarify the role of RLS as a health determinant.

## Methods

Using a cross-sectional study design, we analyzed data from the National Agricultural Workers Survey (NAWS), a nationally representative survey of U.S. farmworkers, and the National Institute of Occupational Safety and Health (NIOSH) Mental Health Supplement. The NAWS is an annual employment-based probability sample survey of farmworkers sponsored by the U.S. Department of Labor. In 2009 and 2010, NIOSH included a Mental Health Supplement to capture the prevalence and predictors of poor mental health symptoms among farmworkers. As such, we analyzed data from 2009 and 2010 only. The NAWS and NIOSH Mental Health Supplement data are publicly available from the U.S. Department of Labor Employment and Training Administration.

### Participants

A total of 3,691 farmworkers participated in the 2009–2010 NAWS and NIOSH Mental Health Supplement. Eligible individuals were employed in crop agriculture and worked at least four hours over the 15 days before the survey date. The sample was comprised of farmworkers aged 14–81 years old across six geographic regions, including California (29.9%), Northwest (18.2%), Midwest (19.5%), Southwest (7.1%), Southeast (12.5%), and East (12.7%).

### Data collection

The NAWS staff conducted face-to-face interviews with farmworkers at their workplace in English or Spanish. Written informed consent was obtained from each farmworker, and

participants were paid $20 cash for their time. Interviews lasted an average of 60 minutes. The NAWS utilized multistage sampling to account for regional and seasonal fluctuations in agriculture. The sampling year was divided into three seasonal interviewing cycles in February, June, and October. The stages of sampling included geographic region, county or farm labor area, employer, and participant. At the employer sampling stage, a simple random sample of farmworkers was selected. The number of interviews conducted at each site was proportional to the number of farmworkers employed. During the 2009–2010 NAWS, there was a 66% response rate from employers recruited and a 92% response rate from farmworkers agreeing to be interviewed [18].

## Measures

Since farmworkers have various linguistic and cultural backgrounds, the NAWS and NIOSH Mental Health Supplement have been pilot tested to evaluate the appropriateness, internal reliability, and validity of the questionnaire items for their use in the target population [19]. Independent and predictor variables were drawn from the NAWS, and the dependent variable was obtained from the NIOSH Mental Health Supplement.

The dependent variable, self-reported health (SRH), is known to be a valid indicator of general health and a robust predictor of significant health events, including all-cause mortality across diverse cultural and demographic populations and communities [20–22]. SRH was assessed by asking individuals to categorize their health status as excellent, good, fair, poor, or don't know. Most participants rated their health as excellent (20.24%), good (57.32%), or fair (21.98%). The number of individuals who rated their health as poor (n = 14, 0.38%) or don't know (n = 3, 0.08%) was too small to contribute statistically to our analyses. Excellent and good responses were collapsed into a single category, and fair and poor responses were collapsed into a single category. Don't know responses were considered missing. The original measure has good construct validity, internal consistency, and test-retest reliability [19].

Race, ethnicity, and legal status were key independent variables. To assess participant race, participants were prompted, "which of the following do you consider yourself? White; Black or African American; Asian; American Indian or Alaskan Native; Native Hawaiian or Pacific Islander; or other." Ethnicity was assessed by asking participants, "which of the following describes you? Mexican-American; Mexican; Chicano; other Hispanic; Puerto Rican; or not Hispanic or Latino." Racial and ethnic identities were organized into three distinct categories: Hispanic, non-Hispanic non-White, and non-Hispanic White. Legal status was assessed directly by asking, "what is your current legal status in the U.S? U.S. citizen by birth, naturalized U.S. citizen, permanent resident/green card, border crossing/commuter card, pending status, undocumented, temporary resident, or other." Participants were also asked if they had "general work authorization" with yes, no, and don't know response options. Participants were categorized as unauthorized immigrants if they were not citizens and did not have work authorization, authorized immigrants if they were not citizens and did have work authorization, or U.S. citizens.

Farmworker participation in SNAP and WIC was captured by their response to the prompt, "within the last two years, has anyone in your household received benefits from or used the services of any of the following programs?" with yes or no response options for "food stamps" and WIC. Farmworkers were categorized by participation in SNAP or WIC or no participation in SNAP or WIC. Participant demographic characteristics were self-reported, including age, sex, country of birth, marital status, number and age of children in the home, and education level. Participants were categorized as migrant workers if they reported travel to engage in farm work in the 12 months before their interview. Participants who reported a diagnosis of

one or more chronic health conditions (asthma, diabetes, hypertension, tuberculosis, heart disease, urinary tract infections, or an unspecified condition) were grouped for analysis. Family poverty level was defined using farmworker income and household size to estimate family income. Families were considered below the poverty level if family income fell below the federal poverty level in 2009 or 2010. The location of the interview determined the sampling region.

## Analysis

Analysis included cross-tabulations with Chi-square tests of independence and multinomial logistic regressions using SAS statistical software (version 9.4, SAS Institute, Cary, NC, 2016). Chi-square tests allowed analysis of between-group differences in SRH by farmworker characteristics. To examine the associations between SRH, race/ethnicity, and legal status, unadjusted multinomial logistic regression models for SRH and race/ethnicity and SRH and legal status were analyzed. Multinomial logistic regression models were adjusted to include all predictor variables (see Table 1 for complete list) and, using backward elimination retained only significant predictors ($p < 0.05$) and those that were important within the study (those variables that may impact eligibility) to obtain the most parsimonious estimates. To examine associations between SNAP/WIC participation, SRH, and race/ethnicity the adjusted model was stratified by farmworkers who did participate in SNAP/WIC and farmworkers who did not participate in SNAP/WIC. Final multinomial logistic regressions generated unadjusted and adjusted odds ratios of SNAP/WIC participation by race/ethnicity. The NAWS composite weight variable was used for descriptive frequencies and multinomial regression models. The composite weight uses sampling, non-response, and post-sampling factors and adjusts for the number of days worked per week, season, and sampling region to construct weights for calculating unbiased population estimates [23].

As the current study utilizes fully anonymized and publicly available data, this research was approved as exempt by the Oregon State University Institutional Review Board.

## Results

Descriptive statistics on selected variables by SRH are presented in Table 1. Participants were predominantly Hispanic (83.5%), male (76.0%), and between the ages of 26–35 years old (26.9%); had less than a high school education (53.4%); and had no known chronic health conditions (80.4%). Most lived with their spouse (48.7%) and without kids (70.5%). Approximately half (51.7%) were unauthorized immigrants. Although nearly a third of farmworkers lived below poverty level, only 12.2% reported SNAP participation, and 18.0% participated in WIC. More unauthorized farmworkers participated in WIC (12.3%) than SNAP (8.2%). Among authorized non-citizen farmworkers, 3.5% used WIC and 1.8% used SNAP. Among citizen farmworkers, 2.2% used WIC and the same percent used SNAP. Few (7.1%) farmworkers reported participation in both SNAP and WIC.

There were significant differences in SRH by race/ethnicity, legal status, age, education, presence of chronic health conditions, marital status, family poverty status, participation in WIC, and region. Among all race/ethnicity and legal status categories, more farmworkers reported excellent/good SRH compared to fair/poor SRH. Non-Hispanic White and citizen farmworkers more frequently reported excellent/good SRH compared to Hispanic, non-Hispanic non-White, authorized immigrant, and unauthorized immigrant farmworkers. There were no statistically significant differences by sex, number or age of children in the household, migration status, or SNAP participation.

**Table 1. Characteristics of U.S. farmworkers by self-reported health status: National Agricultural Workers Survey, 2009/2010 (n = 3691).**

| | Total Column % (n)[a] | Excellent or Good Column % (n)[a] | Fair or Poor Column % (n)[a] | p-value[b] |
|---|---|---|---|---|
| Race/ethnicity | | | | p<0.0001 |
| Hispanic, all races | 83.5% (3081) | 82.0% (2345) | 88.9% (736) | |
| Non-Hispanic non-White | 3.9% (145) | 3.7% (107) | 4.6% (38) | |
| Non-Hispanic White | 12.6% (463) | 14.3% (409) | 6.5% (54) | |
| Legal status | | | | p = 0.0006 |
| Unauthorized immigrant | 51.7% (2055) | 52.3% (1617) | 49.5% (438) | |
| Authorized immigrant | 20.7% (820) | 19.4% (598) | 25.2% (223) | |
| U.S. citizen | 27.6% (1098) | 28.3% (875) | 25.3% (223) | |
| Age (range), years | | | | p<0.0001 |
| 18–25 | 23.0% (921) | 25.0% (775) | 16.3% (147) | |
| 26–35 | 26.9% (1076) | 27.3% (847) | 25.5% (230) | |
| 36–45 | 22.8% (914) | 22.2% (690) | 24.9% (224) | |
| 46–59 | 20.2% (810) | 18.1% (562) | 27.6% (249) | |
| 60+ | 5.4% (218) | 5.5% (170) | 5.3% (48) | |
| Sex | | | | |
| Female | 24.0% (961) | 23.1% (718) | 27.0% (243) | p = 0.02 |
| Male | 76.0% (3042) | 76.9% (2386) | 73.0% (656) | |
| Education | | | | |
| Primary school or less[c] | 53.4% (2138) | 49.6% (1159) | 66.5% (598) | p<0.0001 |
| Beyond primary school | 46.6% (1866) | 50.4% (1564) | 33.5% (301) | |
| Chronic health conditions[d] | | | | |
| None | 80.4% (3217) | 84.5% (2624) | 66.0% (594) | p<0.0001 |
| One or more chronic conditions | 19.6% (786) | 15.5% (480) | 34.0% (306) | |
| Marital status | | | | |
| Single | 36.4% (1456) | 38.3% (1187) | 29.9% (269) | p<0.0001 |
| Married, living with spouse | 48.7% (1950) | 47.6% (1476) | 52.6% (473) | |
| Married, not living with spouse | 12.9% (516) | 12.0% (373) | 16.0% (143) | |
| Other | 2.0% (80) | 2.1% (66) | 1.5% (14) | |
| Children in household | | | | |
| No children | 70.5% (2340) | 70.5% (1807) | 70.7% (532) | p = 0.15 |
| Children under 6 | 17.1% (567) | 17.5% (450) | 15.6% (117) | |
| Children aged 6–17 | 8.5% (283) | 8.0% (206) | 10.3% (78) | |
| Children under 6 and aged 6–17 | 3.8% (127) | 3.9% (101) | 3.4% (26) | |
| Family poverty | | | | |
| Below poverty level | 31.0% (1237) | 32.3% (1001) | 26.5% (236) | p = 0.001 |
| At or above poverty level | 69.0% (2753) | 67.7% (2099) | 73.5% (654) | |
| Migrant | | | | |
| Does not travel for work | 72.7% (2908) | 73.1% (2269) | 71.0% (639) | p = 0.20 |
| Travels for work | 27.4% (1095) | 26.9% (834) | 29.0% (261) | |
| SNAP | | | | |
| Participates in SNAP | 12.2% (486) | 12.3% (380) | 11.8% (106) | p = 0.70 |
| Does not participate in SNAP | 87.9% (3517) | 86.9% (2723) | 88.2% (794) | |
| WIC | | | | |
| Participates in WIC | 18.0% (720) | 19.2% (596) | 13.8% (124) | p = 0.0002 |
| Does not participate in WIC | 82.0% (3284) | 80.8% (2508) | 86.2% (776) | |

*(Continued)*

**Table 1.** (Continued)

| | Total Column % (n)[a] | Excellent or Good Column % (n)[a] | Fair or Poor Column % (n)[a] | p-value[b] |
|---|---|---|---|---|
| Sampling region | | | | p = 0.0002 |
| East | 12.7% (510) | 12.9% (400) | 12.1% (109) | |
| Southeast | 12.5% (500) | 12.7% (393) | 11.9% (107) | |
| Midwest | 19.5% (781) | 20.5% (643) | 15.3% (138) | |
| Southwest | 7.1% (286) | 6.5% (201) | 9.4% (85) | |
| Northwest | 18.2% (730) | 17.4% (541) | 21.0% (189) | |
| California | 29.9% (1197) | 29.9% (925) | 30.2% (272) | |

[a] Columns show weighted sample sizes and frequencies

[b] P-values reflect the significance of Rao-Scott Chi-square between self-reported health and the farmworker characteristic

[c] Primary school or less includes individuals who had no formal schooling and any education through 8th grade

[d] Has been diagnosed with asthma, diabetes, hypertension, tuberculosis, heart disease, urinary tract infections, or unspecified

Table 2 shows the odds ratios (OR) corresponding to SRH by race and ethnicity. Compared to non-Hispanic White farmworkers, those who identified as Hispanic were less likely to report excellent or good health (OR 0.27, 95% CI 0.12–0.62). Similar findings were estimated for non-Hispanic non-White farmworkers, who were less likely to report excellent or good

**Table 2. Odds of self-reporting excellent or good health vs. fair or poor health among U.S. farmworkers by race and ethnicity: National Agricultural Workers Survey (NAWS), 2009/2010 (n = 3691).**

| | Unadjusted Odds Ratio[b] (95% CI) | Adjusted[a] Odds Ratio[b] (95% CI) |
|---|---|---|
| | Excellent or Good Health | Excellent or Good Health |
| **Race/ethnicity** | | |
| Hispanic, all races | 0.41** (0.24–0.71) | 0.27** (0.12–0.62) |
| Non-Hispanic non-White | 0.27*** (0.13–0.58) | 0.27** (0.12–0.59) |
| Non-Hispanic White | 1.0 | 1.0 |
| **Legal status** | | |
| Unauthorized immigrant | | 0.97 (0.48–1.97) |
| Authorized immigrant | | 1.17 (0.56–2.44) |
| U.S. citizen | | 1.0 |
| **Chronic health conditions[c]** | | |
| One or more chronic condition | | 0.34*** (0.23–0.49) |
| None | | 1.0 |
| **SNAP** | | |
| Participates in SNAP | | 0.81 (0.47–1.39) |
| Does not participate in SNAP | | 1.0 |
| **WIC** | | |
| Participates in WIC | | 1.71** (1.10–2.67) |
| Does not participate in WIC | | 1.0 |

[a] Adjusted for legal status, chronic health conditions, use of SNAP, and use of WIC

[b] P-values reflect the significance of the association between self-reported health and race/ethnicity

*p<0.05

**p<0.01

***p<0.001

[c] Has been diagnosed with asthma, diabetes, hypertension, tuberculosis, heart disease, urinary tract infections, or unspecified

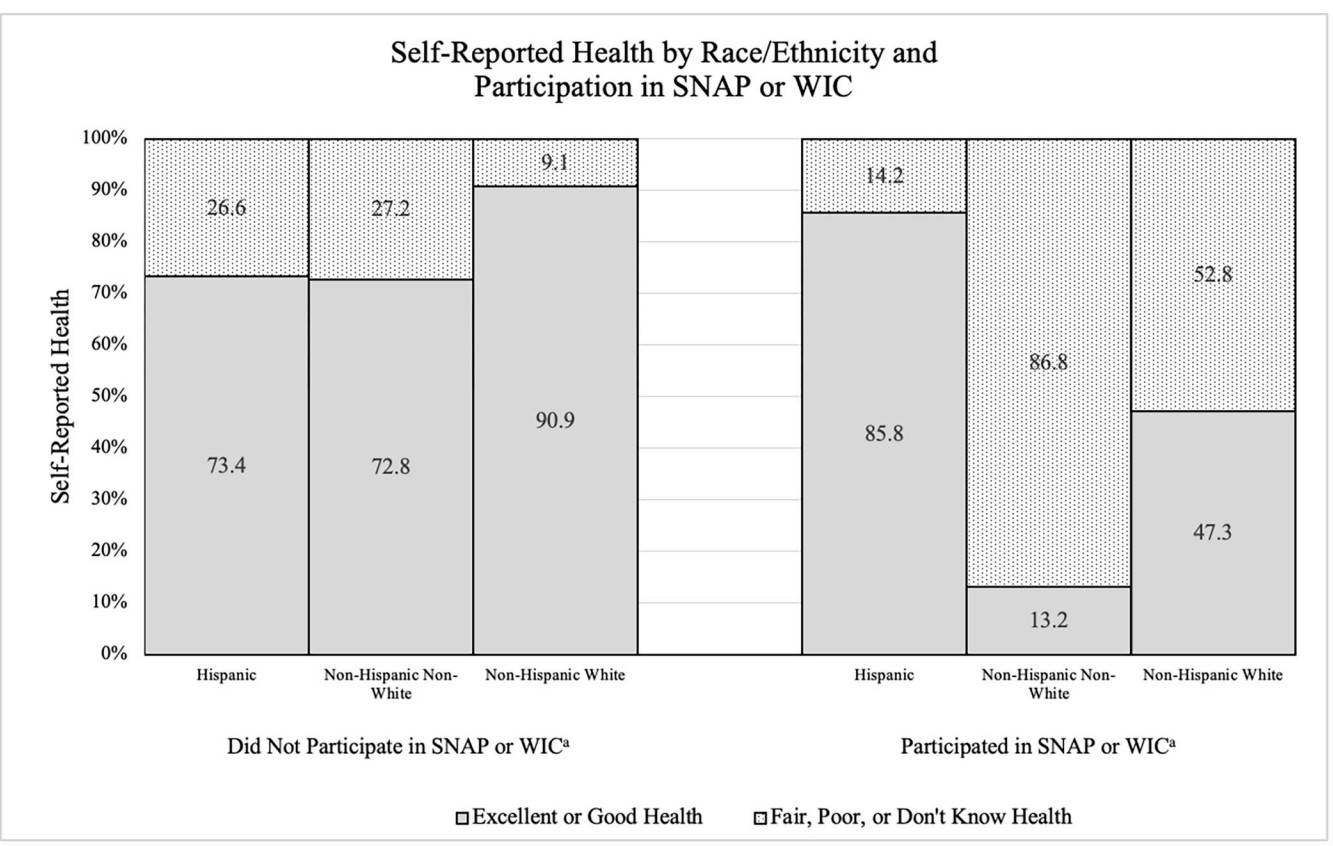

**Fig 1. U.S. farmworker self-reported health status by race/ethnicity and SNAP or WIC participation: National Agricultural Workers Survey, 2009/2010 (n = 3691).** [a] Indicated use of SNAP or WIC within two years prior.

SRH (OR: 0.27, 95% CI 0.12–0.59) than non-Hispanic White farmworkers. Multinomial logistic regression models examining SRH by legal status revealed no differences among unauthorized immigrants (OR 0.97, 95% CI 0.48–1.97) and authorized immigrants (OR 1.17, 95% CI 0.56–2.44) compared to U.S. citizens.

Fig 1 illustrates the relationship between SRH by race/ethnicity and the use of SNAP or WIC. When the sample was stratified by participation in SNAP or WIC, there was significant effect modification (p<0.001) in the model (Table 3). Among farmworkers who did not participate in SNAP or WIC, we found racial/ethnic trends consistent with the non-stratified model (Table 2). Participation in SNAP or WIC modified this association. After adjusting for significant predictor variables, the model produced greater odds of reporting excellent or good SRH (OR 6.74, 95% CI 1.54–29.57) among Hispanic farmworkers who participated in SNAP or WIC compared to non-Hispanic White farmworkers. Among SNAP/WIC participants, unauthorized immigrant farmworkers had lower odds of reporting excellent or good health compared to citizen farmworkers (OR 0.19, 95% CI 0.05–0.78). No significant differences were found for non-Hispanic, non-White farmworkers or authorized immigrant farmworkers who participated in SNAP or WIC.

## Discussion

This study provided two main findings regarding RLS as a social determinant of farmworker health. First, results confirmed disparities in farmworker SRH by race/ethnicity but not by

**Table 3. Odds of self-reporting excellent or good health vs. fair or poor among U.S. farmworkers by race and ethnicity, stratified by participation in SNAP or WIC[a]: National Agricultural Workers Survey (NAWS), 2009/2010 (n = 3691).**

| | Unadjusted Odds Ratio[c] (95% CI) | Adjusted[b] Odds Ratio[c] (95% CI) |
|---|---|---|
| | Excellent or Good Health | Excellent or Good Health |
| **Participated in SNAP or WIC[a]** | | |
| Race/ethnicity | | |
| Hispanic, all races | 5.66** (1.61–19.90) | 6.74** (1.54–29.57) |
| Non-Hispanic non-White | 1.62 (0.11–24.37) | 0.21 (0.03–1.69) |
| Non-Hispanic White | 1.0 | 1.0 |
| Legal status | | |
| Unauthorized immigrant | | 0.19* (0.05–0.78) |
| Authorized immigrant | | 0.31 (0.10–1.00) |
| U.S. citizen | | 1.0 |
| Chronic health conditions[e] | | |
| One or more chronic condition | | 0.47* (0.13–1.12) |
| None | | 1.0 |
| **Did not participate in SNAP or WIC[a]** | | |
| Race/ethnicity | | |
| Hispanic, all races | 0.19*** (0.11–0.35) | 0.19*** (0.08–0.45) |
| Non-Hispanic non-White | 0.17*** (0.07–0.44) | 0.26** (0.12–0.58) |
| Non-Hispanic White | 1.0 | 1.0 |
| Legal status | | |
| Unauthorized immigrant | | 1.02 (0.48–2.21) |
| Authorized immigrant | | 1.16 (0.52–2.59) |
| U.S. citizen | | 1.0 |
| Chronic health conditions[e] | | |
| One or more chronic condition | | 0.31*** (0.21–0.48) |
| None | | 1.0 |

[a] Indicated use of SNAP or WIC within two years prior

[b] Adjusted for legal status and chronic health conditions

[c] P-values reflect the significance of the association between self-reported health and race ethnicity

*$p<0.05$

**$p<0.01$

***$p<0.001$

[e] Has been diagnosed with asthma, diabetes, hypertension, tuberculosis, heart disease, urinary tract infections, or unspecified

legal status. Second, participation in SNAP or WIC may modify the negative association between SRH and race/ethnicity.

Consistent with previous studies on farmworker health disparities, Hispanic and non-Hispanic non-White farmworkers had a lower likelihood of reporting excellent or good health than non-Hispanic White farmworkers [18, 22]. Still, there was no overall association between SRH and legal status. These findings suggest that race and ethnicity may be more salient constructs associated with excellent and good SRH ratings among farmworkers. As such, RLS may not shape farmworker health through primary pathways that directly impact the health of those with discredited legal status.

The absence of any significant association between SRH and legal status identified in the current study adds to the limited body of research examining legal status and health, which

offers mixed evidence [5, 13, 24]. One meta-analysis of 40 studies showed direct relationships between both unauthorized legal status and poor mental health outcomes like depression, anxiety, and post-traumatic stress disorder and also between unauthorized legal status and suppressed access to health services [13]. Other studies suggest that unauthorized immigrants may experience better physical health than authorized immigrants [5]. Such contradictions may be related to the Latinx health paradox [25] which describes the relative health advantage observed among Latinx populations, including those with unauthorized legal status, despite social and economic disadvantage. There are several explanations for this phenomenon both generally and among farmworkers [26], including self-selection of immigrants healthy enough to immigrate and engage in physical farm labor [5] and cultural and social protective factors that may taper over time due to acculturation, cumulative disadvantage, and stress [25, 26]. Another explanation posits that the Latinx health paradox is simply a consequence of methodological limitations, including the use of proxy measures of legal status and health [18], the use of inconsistent definitions of Latinx identity [25], small, non-representative sample sizes [27], and measurement errors as a result of underreporting of unauthorized status and health problems and undercounting deaths [25]. Due to such limitations, it may be difficult to study the true relationship between RLS and SRH among primarily Latinx populations like farmworkers.

We found that the Hispanic farmworkers who indicated participation in SNAP or WIC were more likely to report excellent or good SRH than non-Hispanic White farmworkers using the same programs. These findings are consistent with literature that documents the positive roles that SNAP and WIC play in health promotion and their impact on SRH [28] and demonstrates the potential to target farmworker health through enhanced access to these safety net programs. Though SNAP and WIC are ostensibly available to all eligible individuals, underutilization persists among distinctly vulnerable subgroups, including Latinx immigrants and, more specifically, farmworkers [16, 29]. The 12.2% SNAP and 18.0% WIC participation rates measured in this study are on the low end of recent estimates for U.S. farmworkers, ranging from 15–32% for SNAP and 13–22% for WIC [29]. Several studies using the NAWS have found that legal status has little direct association with SNAP and WIC participation rates among farmworkers [30–32]. Still, it may be challenging to capture unbiased SNAP/WIC participation. Farmworkers who openly disclose unauthorized status may experience less mistrust and be more likely to utilize and report utilization of public assistance.

Participation in SNAP and WIC may be pathways through which RLS determines Hispanic farmworkers' health. As demonstrated in Asad and Clair's conceptual model, policies that restrict the use of public assistance by legal status may impact racial/ethnic groups with social proximity to those with discredited legal status through community-level shifts in social and cultural norms, stigma, policy acceptance, and knowledge of available assistance [33]. Many barriers that limit SNAP and WIC participation, including misconceptions about program eligibility and anxieties about the threats imposed by immigration enforcement and anti-immigrant rhetoric, may be reasonably attributed to RLS [33–36]. Notably, the public charge rule, which included immigrants' use of public benefits in determinations of visa renewal and permanent residency in the U.S., deters participation in SNAP and WIC. Recent changes to the public charge rule, widespread confusion and mistrust, and misinformation regarding which programs would be affected have resulted in many immigrants, including lawful permanent residents, terminating their benefits rather than threatening their own or family members' legal status [37–40].

This research has some limitations. First, the cross-sectional study design limits the ability to determine causality. Second, as the NAWS sampling methods rely on employer permission to recruit and conduct interviews on-site, farms in compliance with occupational health mandates may have been more likely to host NAWS researchers. Thus, farmworkers in this study may have had lower occupational health risks than farmworkers on non-compliant farms.

Further, the variables examined in this study were based on self-reported data collected on-site, which may have undermined internal validity. For example, participants might have over-rated their health if their employers or supervisors were present. Next, as the NIOSH Mental Health Supplement was only collected over two years, we only had SRH data from 2009/2010, limiting the sample size and restricting our ability to explore changes in RLS and SRH coinciding with shifts in policy. Lastly, as the sample size of non-Hispanic White SNAP/WIC participants was small (n = 27), we note that the role of SNAP/WIC as an effect modifier should be interpreted with caution, as estimates may be unstable.

The measure of health included in the current study, SRH, is a proxy measure based upon participants' subjective interpretations of health. While SRH, health, and chronic disease are related factors [20–22] discussed throughout this paper, it is important to note that they are distinct constructs. SRH is dependent upon participants' subjective interpretations of the health they experience. In contrast, the term "health" generically represents a state of being free from illness or injury but is also subject to interpretation. Chronic disease refers to persistent conditions or diseases that often require a clinical diagnosis and ongoing medical attention, such as diabetes, hypertension, or heart disease. In contrast, SRH is dependent upon participants' subjective interpretations of the health they experience. How health is embodied and understood may vary by cultural context, which may have affected the ways that farmworkers responded to the SRH measure.

Future studies on RLS as a social determinant of farmworker health using the NAWS would be strengthened with an amended survey that includes SRH as a standard measure to be collected annually. It is critical that the value of SNAP and WIC participation is examined among a broader sample of farmworkers. Cluster sampling to oversample farmworkers who participate in SNAP or WIC may clarify the relationship between race/ethnicity, SRH, and program participation. We can generate more stable estimates by purposefully balancing the sample across demographic characteristics, particularly racial and ethnic groups, to draw accurate comparisons. Longitudinal studies that explore farmworker SRH before and during SNAP or WIC participation may strengthen our understanding of the role SNAP and WIC play in overall health. Additional research is needed to explore how access to and use of health-promoting public programs and resources function as pathways through which RLS determines farmworker health.

## Conclusions

This study explored differences in SRH by individual and household characteristics in a nationally representative sample of farmworkers, a predominantly Hispanic immigrant workforce. Study findings provide valuable information regarding the relationships among legal status, race, ethnicity, health, and public assistance participation. The racial and ethnic health disparities in farmworker SRH found in this study suggest that legal status has no direct association with farmworker SRH. Evidence of effect modification by SNAP and WIC participation highlights the potential to improve farmworker SRH by expanding access to and utilization of existing resources. Hispanic farmworkers who rated their health as fair or poor may derive the most benefit from participation in SNAP and WIC. These findings underscore the need to expand U.S. farmworkers' eligibility for and facilitate participation in SNAP and WIC and to consider RLS as a fundamental social determinant of health among farmworkers.

## Acknowledgments

We would like to thank Marc Braverman for his comments on later drafts of the article.

## Author Contributions

**Conceptualization:** Briana E. Rockler.

**Formal analysis:** Briana E. Rockler, Ellen Smit.

**Investigation:** Briana E. Rockler, Ellen Smit.

**Methodology:** Briana E. Rockler, Ellen Smit.

**Resources:** Briana E. Rockler.

**Supervision:** Stephanie K. Grutzmacher.

**Visualization:** Briana E. Rockler.

**Writing – original draft:** Briana E. Rockler.

**Writing – review & editing:** Briana E. Rockler, Stephanie K. Grutzmacher, Jonathan Garcia, Ellen Smit.

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
