## [Decision Letter · Decision Letter 0]

16 May 2022

PONE-D-21-36136The role of SNAP and WIC participation and racialized legal status in U.S. farmworker healthPLOS ONE

Dear Dr. Rockler,

Thank you for submitting your manuscript to PLOS ONE. After careful consideration, we feel that it has merit but does not fully meet PLOS ONE’s publication criteria as it currently stands. Therefore, we invite you to submit a revised version of the manuscript that addresses the points raised during the review process. Your manuscript has been evaluated by one reviewer whose comments can be found below. To ensure that your manuscript meets our fourth publication criterion please address the reviewer's recommendation with respect to the presentation of your results and references to statistical significance. In addition, for clarity purposes please ensure that any acronyms (eg. SNAP) are spelt out in the abstract.

We look forward to receiving your revised manuscript.

Kind regards,

Alejandra Clark

Division Editor

PLOS ONE

**Journal Requirements:**

**Reviewers' comments:**

Reviewer's Responses to Questions

**Comments to the Author**

1. Is the manuscript technically sound, and do the data support the conclusions?

Reviewer #1: Yes

2. Has the statistical analysis been performed appropriately and rigorously? 

Reviewer #1: Yes

3. Have the authors made all data underlying the findings in their manuscript fully available?

Reviewer #1: Yes

4. Is the manuscript presented in an intelligible fashion and written in standard English?

Reviewer #1: Yes

5. Review Comments to the Author

Reviewer #1: This is a very interesting article, addressing an issue of great importance. The analyses are performed with rigor, and the conclusions are not over-stated.

One minor area of concern is that the authors use the term "approached signfiicance" or near-significant (p=0.07) describing the relationship between self-reported health status and legal immigration status. It would be more accurate to say simply that this relationship was not significant. Because of the multiple variables considered in table 1, one could argue that even p< 0.05 is a generous threshold for significance, given the problem of multiple comparisons.

I also found the tables to be somewhat difficult to interpret. Fair and poor were collapsed due to sample size issues, but I could see collapsing Excellent and Good into one category (dichotomizing the SRH into excellent / good vs fair/poor) would make sense and be more interpretable.

The sentence (lines 276-280) in the discussion where the authors wrestle with the lack of association between SRH and legal status is confusing and perhaps self-contradictory, in part because the authors seem committed to making the case that the study results confirm their conceptual framework of racialized legal status as a predictor of health outcomes. In contrast, the sentence (lines 283-284) is quite clear, succinct, and represents the actual findings that

a) there are significant racial-ethnic differences in SRH

b) there are no significant differences in SRH by legal status

c) participation in SNAP/WIC was associated with significantly better SRH in Hispanic farmworkers.

6. PLOS authors have the option to publish the peer review history of their article (what does this mean?). If published, this will include your full peer review and any attached files.

Reviewer #1: No

---

## [Author Response · Author response to Decision Letter 0]

30 Jun 2022

This manuscript was returned after review with a request for minor revisions. We are grateful for the reviewer's feedback and have integrated their recommendations. We believe the article has improved as a result of these changes. Specific responses to the reviewer and editor's comments are included in the table (pages 2-3) that follows the "response to reviewers" document.

---

## [Editor Report · Decision Letter 1]

29 Jul 2022

The role of SNAP and WIC participation and racialized legal status in U.S. farmworker health

PONE-D-21-36136R1

Dear Dr. Rockler,

We’re pleased to inform you that your manuscript has been judged scientifically suitable for publication and will be formally accepted for publication once it meets all outstanding technical requirements.

Kind regards,

Edward Jay Trapido, ScD

Academic Editor

PLOS ONE
---

## [Editor Report · Acceptance letter]

8 Aug 2022

PONE-D-21-36136R1 

The role of SNAP and WIC participation and racialized legal status in U.S. farmworker health 

Dear Dr. Rockler:

I'm pleased to inform you that your manuscript has been deemed suitable for publication in PLOS ONE. Congratulations! Your manuscript is now with our production department. 

Kind regards, 

on behalf of

Dr. Edward Jay Trapido 

Academic Editor

PLOS ONE